# Do Historic Landscape Images Predict Tourists' Spatio-Temporal Behavior at Heritage Sites? A Case Study of West Lake in Hangzhou, China

Qiqi Liu [1], Xiaolan Tang [1,2,*] and Ka Li [1]

1   College of Landscape Architecture, Nanjing Forestry University, Nanjing 210037, China
2   NFU Academy of Chinese Ecological Progress and Forestry Studies, Nanjing Forestry University, Nanjing 210037, China
*   Correspondence: xiaolant@njfu.edu.cn

**Abstract:** Rich in history and culture, heritage sites often evoke stirring emotions and memories. We analyzed historical poetry using grounded theory and high-frequency word and semantic analysis to construct historic landscape images (HLIs) of the West Lake UNESCO World Heritage Site in Hangzhou, China. GPS trajectory data were used to identify hot and cold tourist spots and understand the site's intra-attraction tourist behaviors (IATBs). Finally, we analyzed the HLI–IATB relationship. We found that the tourist distribution was uneven, and different attractions had different visitor behaviors. Our findings should inform future heritage site management—and neighboring cities—about the benefits of using HLIs to predict attraction visitors' behaviors and leveraging those insights to optimize multiple-attraction sites proportionally. Such projections can provide new perspectives for heritage studies, landscape planning, and tourism image-making.

**Keywords:** historic landscape images; intra-attraction tourist behaviors; GPS; big data; world heritage sites





## 1. Introduction

Cultural tourism plays a crucial role in social and economic development. Heritage resources increasingly drive cultural tourism [1,2]. Heritage sites can have cultural, historical, scientific, or other significant elements [3]. Most visitors to heritage sites are seeking a "historical heritage experience" [4]. However, their expectations differ widely. Therefore, understanding tourists' preconceptions about the heritage sites they plan to visit is vital for successful heritage management.

Although heritage sites are closely related to history, it is unclear how history might affect future tourism trends. We wondered whether exposure to images of historic landscape images (HLIs) might influence tourists' travel plans, decisions, and behaviors. If so, it could help predict future heritage site tourism trends, facilitate tourism management, provide valuable cultural content for product development and marketing of heritage tourism, and enhance tourism's economic benefits.

This study explored the association between HLIs and intra-attraction tourist behaviors (IATBs), and used HLIs to predict possible future tourist behaviors. This study describes how HLIs could be used to enhance the tourism experience and help heritage tourism management predict visitors' behaviors. We also discuss future considerations and implications.

### 1.1. Destination Images

An *image* is a visual impression or an individual's perception of an environment. The term "image" was introduced by Kevin Lynch (1918), who pioneered studying the elements of urban space in terms of sensory forms in his five-year investigation of how observers absorb information about cities. He emphasized the importance of a city's *legibility* (the ease with which people can discern a city's patterns and landmarks) and *imageability* (how easily

a physical object, word, or environment evokes a mental image) and how images form memory spaces. Images capture an individual's perspective and the interplay between the elements in the overall understanding of a setting [5].

The study of destination images can help us understand the role the images of a city play in attracting tourists [6]. According to some researchers, destination images are made up of both cognitive and affective images [7]; some studies demonstrate the hierarchical integration process of creating a destination image using machine learning to explore destination images and cultural identity from the perspective of linguistic landscapes [8,9]. Other researchers have analyzed destination images from the perspective of textual information through combinatorial analysis [10], using text mining to track changes in travel destination images [11], and the analysis of word frequencies and co-occurrence networks through online travelogues [12].

Heritage landscapes are rich in historical and cultural relics that frequently feature in destination images. Representations of the physical and cultural landscapes evoke emotional responses in the human brain through direct or indirect sensory inputs. Many have a special significance to the beholders derived from historical influences or experiences of nature, culture, and aesthetics. We can obtain HLIs using historical texts to advance our understanding of heritage site elements in modern landscapes.

### 1.2. Tourists' Spatio-Temporal Behaviors

Analyzing tourists' spatio-temporal behaviors has become a popular research topic for scholars in multidisciplinary fields. Tourists' spatio-temporal behaviors include inter-destination and intra-attraction movement and can be measured using various scales [13–15]. The present study examined IATBs.

Initial research on IATBs focused on spatio-temporality and mobility [16,17]. These studies demonstrated the degree of spatial concentration among tourists and time spent in different destinations. Some scholars have studied tourists' movement patterns, reflecting their chosen routes and points of interest with geometric models [18,19]. Tourist movement patterns can be structured by *nodes* (focal points and thematically related clusters of attractions, accommodations, infrastructure, and services) [20] that are connected by *paths* (channels through which people move). This concept follows the environmental elements of Lynch's (1960) city image [5]. The classification of nodes can be used to indicate a destination's attractiveness [21].

Using the geographical concept of spatio-temporal paths, some researchers have interpreted tourists' behavioral patterns by combining quantitative and qualitative methods in terms of temporal behavioral factors, spatial behavioral factors, activity selection factors, and path characteristics [22–24]. Others have shown that destination images influence visitor behaviors [25]. However, the interaction between tourists and attractions during visits is still under-researched [26].

### 1.3. Big Data GPS Technology and Spatio-Temporal Behavior

Studying IATB requires detailed data on visitors' behaviors. However, traditional data acquisition methods (e.g., telephone surveys, on-site surveys, etc.) can be time- and energy-consuming, inaccurate, and limited [27]. Studies of mobility within destinations have gradually increased with the development of new tracking technologies such as the global positioning system (GPS) [28]. Big data GPS technology has many advantages when applied to spatio-temporal behavioral tourist activities. For example, it can objectively and accurately track and record individual tourists' spatio-temporal paths [29], which is nearly impossible using tourists' subjective evaluations [30]. GPS technology and big data analysis enable comparisons between subjective and objective visitor behaviors, broadening the scope of academic investigations and contributing to heritage site optimization.

GPS trackers can record tourists' trajectories as points in space-time, providing precise spatial and temporal data about their trips [31]. Another useful new technological tool in such investigations is geographic information system (GIS) databases. The combination of

GPS and GIS analyses can significantly improve the visualization of tourists' movements as well [32]. Some scholars have collected GPS data using mobile devices and analyzed the clustering of tourists' trajectories to determine their behavior patterns [33]. Shoval (2008) used a 10 × 10 m grid to show tourist trajectory clusters to demonstrate tourists' spatial distribution density at different periods [34]. Summarizing the connections between each part of each visitor's dwell grid can reveal the spatial relationships of the landscape [35].

GPS devices cannot directly collect personal or trip information (e.g., age, gender, trip mode, trip purpose). However, combining GPS data with traditional surveys (e.g., trip/activity diaries, mental or cognitive maps, etc.) can more fully capture detailed behaviors [31]. In China, big data GPS technology has been used to study IATB [36]. Other research has analyzed tourist travel activities and spatio-temporal behavior patterns using data mining heuristics [37], highlighting the effectiveness of this method. These studies show that visualization and clustering studies of IATB based on big data and GPS are essential to investigations of tourists' spatio-temporal behaviors.

This study aimed to understand the relationship between HLIs and IATBs and its implications. We considered three research questions:

1.  How are the HLIs of heritage sites perceived?
2.  How do HLIs explain visitors' spatio-temporal behaviors?
3.  What are the implications of HLIs for the future of heritage tourism management and urban development?

## 2. Materials and Methods

### 2.1. Study Area

West Lake, a UNESCO World Heritage Site, is divided into nine scenic areas with 287 sites on 3322.88 ha. The lake surface is 559.30 ha. The site is surrounded on three sides by mountains and connected to the city on the other. West Lake evolved from a lagoon around 220 BC. Between the 9th and 12th centuries, officials added West Lake's main artificial features, including embankments, hills, temples, pagodas, pavilions, gardens, and trees, and constructed two causeways and three islands (Figure 1). Its best-known points of interest, the "Ten Poetically Named Scenic Places", are named for poems from the mid-13th century and represent an idealized classical landscape with humans and the environment in perfect harmony. By combining rich history and culture with natural beauty, the West Lake heritage site has created a unique landscape image, named the "Poetic and Picturesque Landscape" by UNESCO. West Lake can be seen as a cultural image and a visible and symbolic expression of the human–environment relationship that has developed over thousands of years. The memorable historical associations found in many of its sites, its high visitor traffic, and its visitors' diverse spatio-temporal behavior make the West Lake heritage site an ideal case study.

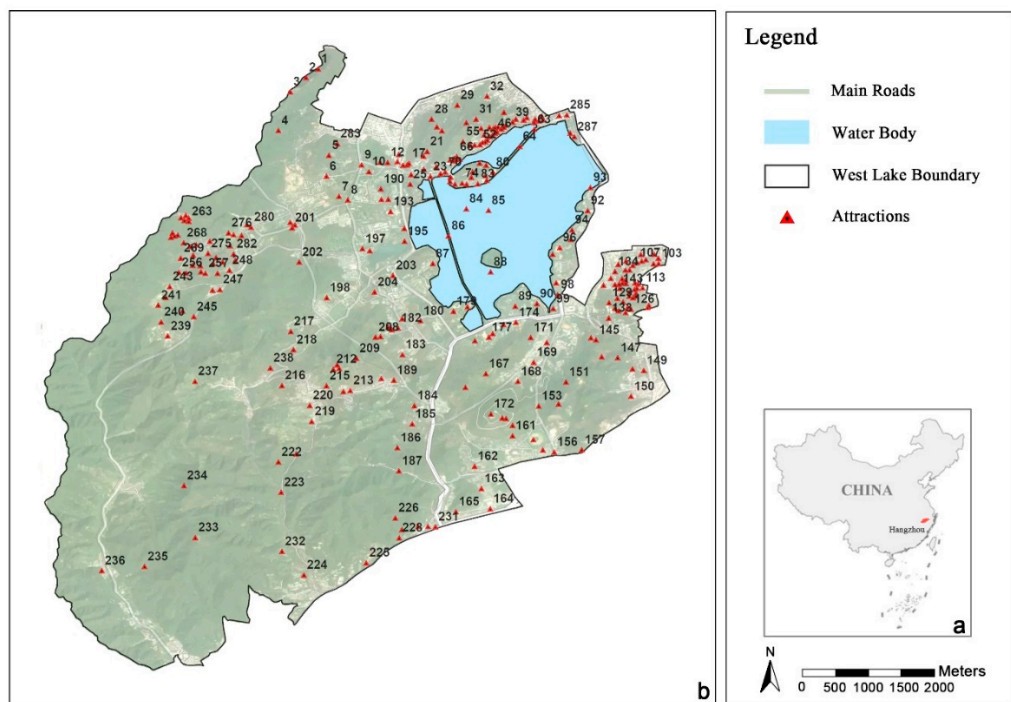

**Figure 1.** Map of West Lake. Subgraph (**a**): location of West Lake in China. Subgraph (**b**): 287 attractions at West Lake.

## 2.2. Analytical Framework

Figure 2 illustrates this study's research framework.

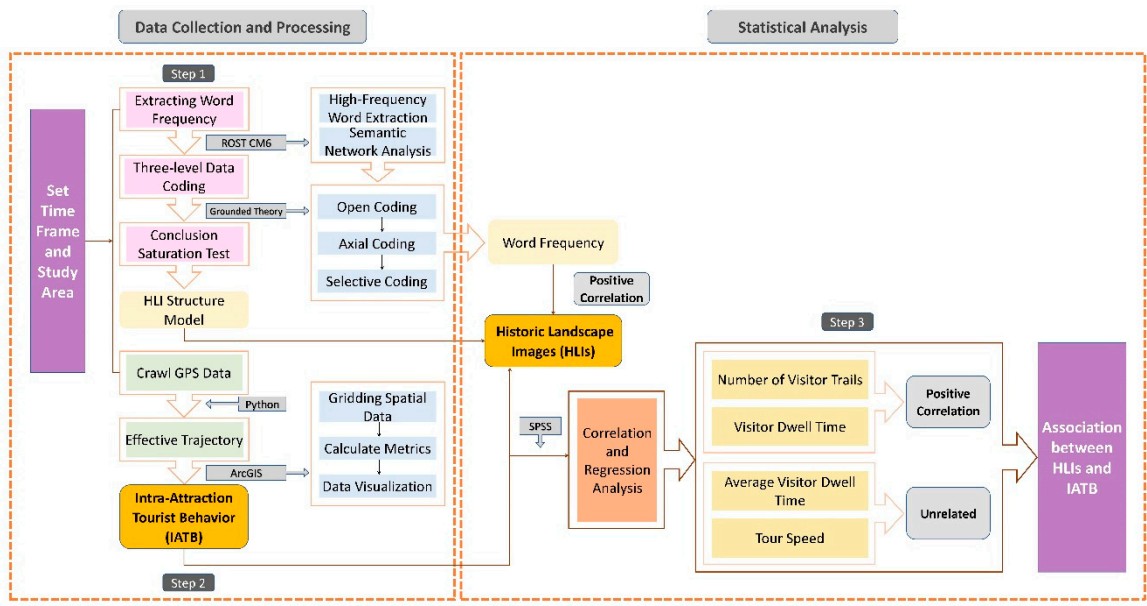

**Figure 2.** Schematic diagram of the research method.

## 2.3. Data Collection

### 2.3.1. Historic Landscape Images

Using grounded theory (see Section 2.4.1) as our framework and relevant ancient poetry as our study material, we defined West Lake's HLIs using the ROST CM6 software, which can perform high-frequency word, semantic network, and sentiment analyses on text passages. Ancient Chinese poems are cognitive expressions that typically use subtle

descriptions of environmental landscapes, culture, and emotions, making them suitable for interpreting historic landscapes. West Lake was built from the Tang to the Qing Dynasty (618–1911); therefore, we collected poems from *All Tang Poems* (Tang Dynasty), *All Song Lyrics* (Song Dynasty), *West Lake Excursions* (Ming Dynasty), and *Selected Poems and Lyrics of West Lake* (Qing Dynasty). The poems mentioned 59 of the heritage site's 287 attractions (Figure 3). Therefore, we considered them relevant for HLIs. We coded the 59 attractions by scenic area from A to I (Table A1).

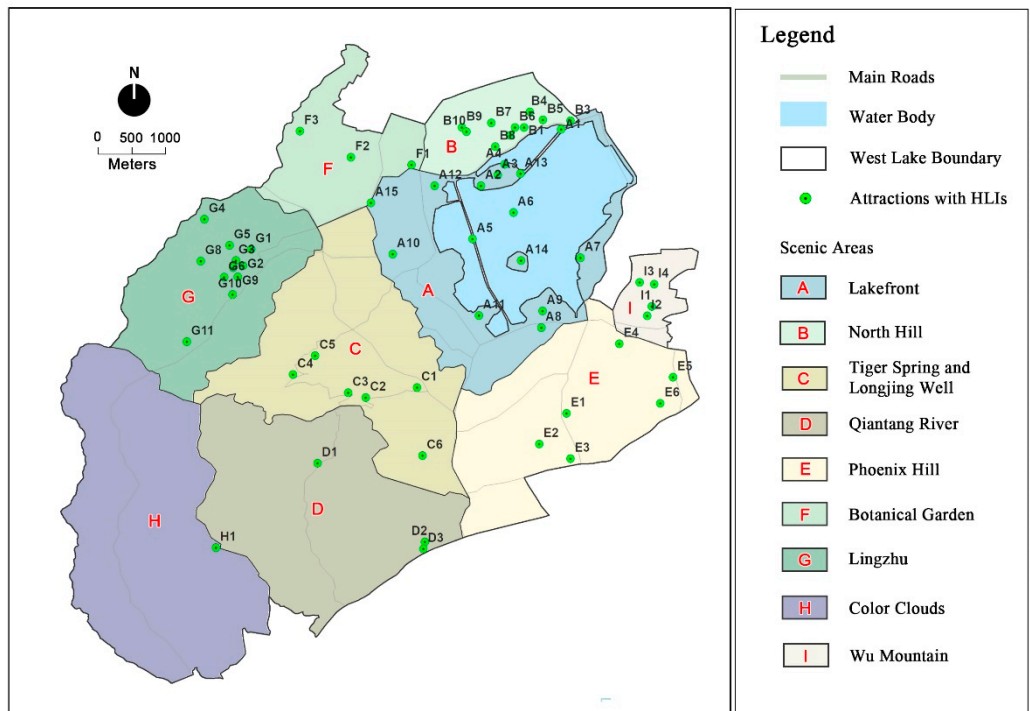

**Figure 3.** Fifty-nine attractions with HLIs in the nine scenic areas of the West Lake heritage site.

We corrected all the texts for obvious errors (content recording errors, punctuation errors, spelling errors, etc.) and deleted duplicate and irrelevant content. We collected 1131 poems, extracted 3993 landscape image terms, and then randomly selected two-thirds of the sample for coding analysis and model construction. We reserved the remaining one-third of the sample for theoretical saturation tests.

The texts were analyzed sentence-by-sentence to ensure validity and rigor and avoid textual bias caused by differences in cognition and expression between historical poetry and modern Chinese. The texts used earlier versions of Chinese, with multiple morphological word variations compared with modern Chinese. Therefore, we first standardized the texts using manual recognition before importing them into the ROST CM6.0 software for word-frequency analysis. The ROST CM6.0 software is a humanities digital research tool developed by Prof. Yang Shen of Wuhan University in China to calculate frequency statistics for text passages and perform cluster and semantic network analyses [38].

We created content clusters for the text—for example, by uniformly replacing the words "flower", "grass", "willow", "peach", "laurel", "cypress", and "lotus" with the word "plant"—for ROST CM6.0 processing. First, we created a custom dictionary of words specific to an attraction. Second, we filtered out irrelevant words such as pronouns and prepositions. Finally, we extracted high-frequency words and analyzed the semantic networks in the text content.

2.3.2. Intra-Attraction Tourist Behaviors

We collected GPS tracking data from 2bulu.com and foooooot.com, China's leading outdoor resource-sharing websites. Web page-recognition rules were created to enable standardized information extraction to handle the complexity of the underlying trajectories when browsing publicly available web pages and stored the visible information in a structured manner.

A Python script was written to crawl the West Lake trajectories voluntarily shared by tourists from March 2019 to September 2021. Despite the reduced number of tourist trips following the COVID-19 outbreak in 2020, we found 1203 datapoints. The appropriate ethics review board approved the study design.

To prevent information errors and clean trajectory data, we designed an extraction-storage scheme rule to isolate valid trajectory information (Figure 4).

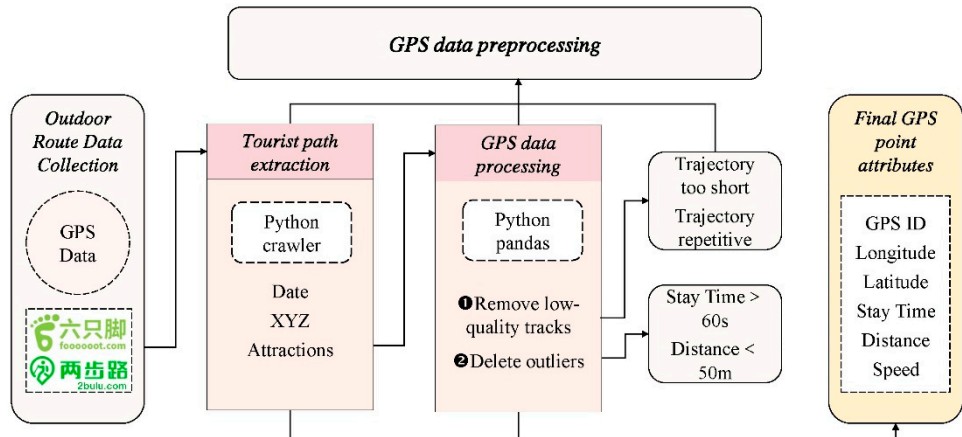

**Figure 4.** Data structure and information extraction-storage scheme rules for valid GPS trajectories.

The first step was removing low-quality trajectories, including those that were too short or repetitive. The second step involved using the Pandas software to calculate the time spent between two points on the same trajectory. The calculations used the absolute value of the difference between two adjacent points, which we obtained as follows:

$$d = \sum_{i=2}^{n} \sqrt[2]{(x_1 - x_{i-1})^2 + (y_i - y_{i-1})^2} \tag{1}$$

The third step was removing outliers. Our field survey of the tourists' behaviors at West Lake using a tachymeter to locate two adjacent trajectory points with the GPS tracking data captured by the network revealed that the time tourists spent between two trajectory points when walking or driving was generally less than 60 s (excepting special circumstances), and the action distance was less than 50 m. Therefore, we considered times spent between two points of more than 60 s or distances between two points of more than 50 m anomalies and removed them. In the fourth step, we imported the processed data into ArcGIS and deleted points outside the study area.

Of the 1203 trajectory datapoints collected, we used 920 valid trajectories for this study, including 1,277,777 valid GPS points (valid percentage 76.4%). Table 1 shows example data. The attribute information for each point included the route number, the sequence number of the point in the route, longitude, latitude, dwell time, the distance between two points, and speed.

**Table 1.** Data Attribute Information.

| Number | N | Longitude | Latitude | Dwell Time | Distance/m | Speed/m/s |
|---|---|---|---|---|---|---|
| 0 | 1 | 117.987288 | 30.004529 | 0 | 0.000 | 0.000 |
| 1 | 1 | 117.987170 | 30.004541 | 12 | 13.136 | 1.095 |
| 2 | 1 | 117.987063 | 30.004610 | 10 | 12.431 | 1.243 |
| 4 | 1 | 117.986925 | 30.004704 | 4 | 3.205 | 0.801 |
| 5 | 1 | 117.986848 | 30.004662 | 18 | 8.838 | 0.491 |
| . . . | . . . | . . . | . . . | . . . | . . . | . . . |
| 241 | 99 | 117.982672 | 30.004618 | 8 | 6.359 | 0.795 |
| 242 | 99 | 117.982701 | 30.004514 | 8 | 6.312 | 0.789 |
| 243 | 99 | 117.982709 | 30.004416 | 14 | 5.190 | 0.371 |
| 244 | 99 | 117.982752 | 30.004322 | 8 | 6.849 | 0.856 |
| 245 | 99 | 117.982786 | 30.004228 | 10 | 6.192 | 0.619 |

Source: http://www.foooooot.com/ and https://www.2bulu.com/ (accessed on 30 September 2021).

*2.4. Data Analysis*

2.4.1. HLI Identification and Construction

We used grounded theory as our framework for analyzing the processed textual materials. Grounded theory is a qualitative research method that systematically summarizes primary materials and constructs substantive theories from the bottom up. The approach encodes the collected material in three formal stages (open coding, axial coding, and selective coding) to find the core categories that reflect the essence of things and construct relevant social concepts [39]. Three-stage coding requires extensive information collection because a theory cannot be formed until there are no more different classes and theoretical saturation and completeness are reached.

The first data coding step in the process of grounded theory research is open coding: the researchers review the textual materials word by word, extract the codes, and perform cluster analysis. After text preprocessing, comparisons, and analyses, we obtained 1131 ancient poetry texts describing West Lake. We proposed 3993 codes for analysis and model construction and generated several initial categories. Table 2 shows some of the original materials and the coding process. Table 3 shows the extracted initial categories and frequencies.

**Table 2.** Original Materials and Coding Process.

| Attractions | Original Materials | Source | Code | Initial Category |
|---|---|---|---|---|
| Lingering Snow on Broken Bridge (A1) | The bridge embankment is lush with smoke and willows, and the dewy grass looks like a skirt. (桥堤烟柳葱菁,露草芊绵,望如裙带) | (Ming Dynasty) *West Lake Excursions* | Bridge, Smoke, Willow, Grass (桥堤,烟,柳,草) | Infrastructure, Smoke, Plants, Vegetation |
| Xiling Printing House (A2) | The mountain is always moist without rain, and the water is clouds free. (不雨山常润,无云水自阴) | (Ming Dynasty) *West Lake Excursions* | Rain, Mountain, Clouds, Water (雨,山,云,水) | Rain, Mountain, Clouds, Lake |
| Three Pools Mirroring the Moon (A14) | The stars are sinking into the river, and the pipes and strings are shining across the river. (入河渐欲沉星斗,隔浦同看照管弦) | (Qing Dynasty) *Selected Poems of West Lake* | River, Stars, Orchestra (河,星斗,管弦) | Rivers, Stars, Feast |
| Twin Peaks Piercing the Cloud (A15) | The high peaks are listed on the distant mountains, and the pavilions have been there for hundreds of years. (高峰列远岑,亭亭几百载) | (Qing Dynasty) *Selected Poems of West Lake* | Peak, Pavilion (峰,亭) | Peaks, Buildings |

**Table 3.** Extracted Initial Categories and Frequencies.

| Initial Category | Frequency | Initial Category | Frequency | Initial Category | Frequency | Initial Category | Frequency |
|---|---|---|---|---|---|---|---|
| Plant | 431 | Peak | 78 | Snow | 30 | Music | 14 |
| Building | 271 | Religion | 67 | Shadow | 30 | Stars | 13 |
| Lake | 235 | River | 65 | Fragrance | 29 | Residence | 12 |
| Mountain | 230 | Boat | 65 | Celebrity | 27 | City | 11 |
| Feast | 203 | Sound | 64 | Dream | 26 | Grave | 11 |
| Fowl | 175 | Spring | 63 | Country | 24 | Farmland | 11 |
| Infrastructure | 167 | Visitor | 60 | Fame | 20 | Valley | 11 |
| Cloud | 158 | Smog | 59 | Wonderland | 19 | Stream | 10 |
| Wind | 145 | Autumn | 58 | Emotion | 17 | Tour | 10 |
| Myth | 144 | Beast | 54 | Fog | 17 | Village | 9 |
| Goods | 123 | Sun | 58 | Palace | 15 | Seclusion | 9 |
| Moon | 101 | Spring | 53 | Cave | 15 | Summer | 8 |
| Rock | 96 | Sea | 49 | Wells | 14 | Food | 8 |
| Waves | 93 | Sky | 47 | Road | 14 | Fire | 6 |
| Rain | 82 | Forest | 40 | Island | 14 | War | 5 |

Axial coding determines the potential logical relationships between categories and uncovers the main ones. We sorted the logical relationships between the codes, grouping the initial categories with similar attributes under the same main category so their contents did not overlap. We defined 17 main categories (Table 4). For example, "Mountain", "Peak", "Valley", and "Cave" fell under Mountain Resources, whereas "Forest", "Plant", "Fowl", and "Beast" were Biological Resources.

**Table 4.** Main HLI categories.

| Number | Main Category | Initial Category |
|---|---|---|
| 1 | Mountain Resources | Mountain, Peak, Valley, Cave |
| 2 | Biological Resources | Forest, Plant, Fowl, Beast |
| 3 | Meteorological Resources | Wind, Smog, Rain, Cloud, Fog, Snow |
| 4 | Astronomical Resources | Sky, Sun, Moon, Stars, Shadow |
| 5 | Water Resources | Lake, Sea, River, Spring, Waves, Stream, Island |
| 6 | Seasonality | Spring, Summer, Autumn |
| 7 | Myth and Legend | Myth, Wonderland |
| 8 | Traditional Chinese Culture | Celebrity, Religion, Custom, Seclusion |
| 9 | Emotional Expression | Emotion |
| 10 | Imagination Association | Dream |
| 11 | Traditional Architecture | Building, Residence, Palace |
| 12 | Infrastructure | Infrastructure, Road |
| 13 | Sightseeing | Feast, Visitor, Tour |
| 14 | Auditory Landscape | Sound, Music |
| 15 | Production and Living | Boat, Goods, Food, Fire, Farmland, City, Grave, Village, Wells |
| 16 | Political Factors | Country, War, Fame |
| 17 | Olfactory Landscape | Fragrance |

Selective coding explores further relationships among the main categories and generates core categories that are overarching, stable, and regular. We organized the 17 main categories into 5 core categories (Table 5). The main categories of Mountain, Biological, Meteorological, Astronomical, Water Resources, and Seasonality comprised the core category of Natural Resources. The Natural Resources category had more frequent mentions than the other categories, making it a core category of West Lake's HLIs. The main categories of Traditional Architecture and Infrastructure fell into the core category of Building Facilities, the second-most important HLI category. The core category of Behavioral Interaction included the main categories of Sightseeing, Production and Living, and Political Factors,

and foregrounded the rich forms of entertainment and diverse daily life experienced at West Lake. The Myth and Legend core category included Myth and Wonderland. The Traditional Chinese Culture core category included Celebrity, Religion, Custom, and Seclusion. These two core categories reflected some of West Lake's unique image and cultural characteristics. Emotional Expression, Imaginative Association, Auditory Landscape, and Olfactory Landscape represented the subjective feelings formed by landscape appreciation and comprised the core category of Aesthetic Activity, an important part of landscape touring. These five core categories together formed West Lake's HLIs.

**Table 5.** Core HLI Categories and subcategories.

| Core Category | Main Category | Frequency |
|---|---|---|
| Natural Resources | Mountain Resources, Biological Resources, Meteorological Resources, Astronomical Resources, Water Resources, Seasonality | 2518 |
| Building Facilities | Traditional Architecture, Infrastructure | 479 |
| Behavioral Interaction | Sightseeing, Production and Living, Political Factors | 580 |
| Cultural Characteristics | Myth and Legend, Traditional Chinese Culture | 266 |
| Aesthetic Activity | Emotional Expression, Imagination Association, Auditory Landscape, Olfactory Landscape | 150 |

We used the remaining one-third of the sample for recoding and categorization to verify that we had achieved theoretical saturation. The results showed that the theoretical categories in the current model had been richly developed. No new theoretical categories and relationships emerged, and no new constituent factors were generated within the five core categories, supporting our theoretical construction of the HLIs.

2.4.2. Construction of IATBs

The construction of IATBs involved four steps (Figure 5).

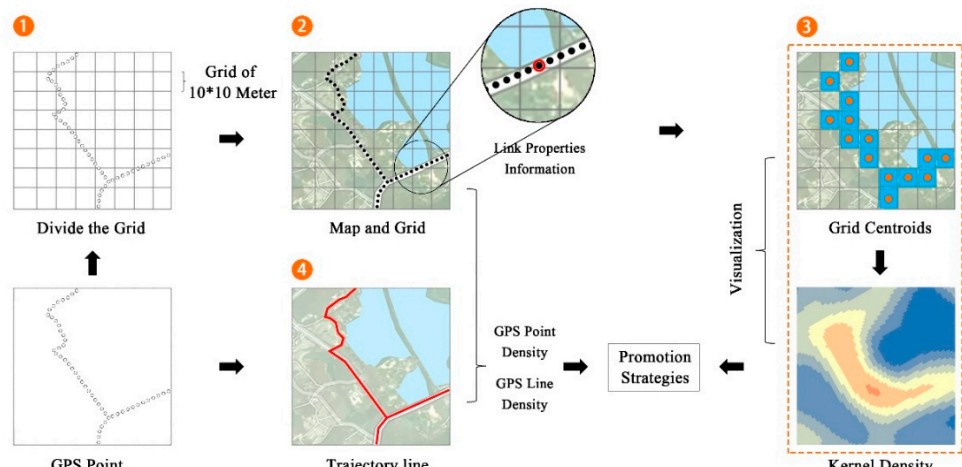

**Figure 5.** Schematic diagram of the construction of IATBs.

First, we placed the spatial data on a grid. Each GPS track comprises a sequence of points with spatio-temporal coordinates. As the information collected from the GPS points did not allow us to determine tourists' dwell status directly, we plotted the spatial data to facilitate visualization and statistics using a specific grid size [40]. Grid statistics rely on GPS track points rather than track lines, allowing for a rapid aggregation of information such as the number of visitors, length of dwell, and geotagging [36]. Converting the spatial data into a grid can significantly reduce errors arising from the influence of a user's mobile device on open GPS data, making it easier to analyze the coupled features and geospatial data [41]. To delineate the trajectories in detail, we set the grid to 10 × 10 m. We used

the distribution of the trajectory points within the grid to determine the tourists' dwell behavior in different spaces.

Second, we linked the data's attribute information. We converted the GPX file of the preprocessed GPS data into elements and imported it into GIS to generate XY coordinates. We then imported the coordinate-converted track data into ArcGIS for extraction and analysis. We imported the attribute information for the location track points and track data into the grid, including the route number, the sequence number of the point in the route, longitude, latitude, dwell time, the distance between two points, and speed information, because these attributes reflected the tourists' behaviors.

Third, we visualized the data's geographic features. To improve the visualization, we used the Feature to Point tool in ArcGIS to extract each grid's centroids to facilitate kernel density and identify hotspots where visitors passed by or congregated, places where the average dwell was longer, and areas of interest to visitors.

Fourth, we used the Points to Line tool in GIS to convert the GPS track points into track lines to determine the number of tracks associated with each attraction. We also generated GPS point density and line density from the GPS track points and lines to produce a precise visualization.

### 2.4.3. Correlation and Regression Analysis

After completing separate analyses of HLIs and IATBs, we combined the correlation and regression analysis results using IBM SPSS Statistics for Windows, Version 23.0 (IBM Corp., Armonk, NY, USA) to gain a deeper understanding of the associations and interactions between the components.

The Spearman correlation coefficient is a nonparametric indicator used to measure the linear correlation between two variables. We used the Kolmogorov–Smirnov (K–S) test to demonstrate that neither variable had a normal distribution. The value of Spearman's correlation coefficient was between −1 and 1. A value of 1 indicates a perfectly positive correlation between two variables; a value of −1 indicates a perfectly negative correlation; and a value of 0 indicates no correlation. We verified the results' significance with the two-tailed *p*-value test. Generally, *p*-values < 0.05 are deemed statistically significant. We calculated the values as follows:

$$\rho = \frac{\sum_i (x_i - \overline{x})(y_i - \overline{y})}{\sqrt{\sum_i (x_i - \overline{x})^2 \sum_i (y_i - \overline{y})^2}} \tag{2}$$

## 3. Results

### 3.1. HLIs and Word Frequency

We ultimately categorized the different HLI codes into five core categories: Natural Resources, Building Facilities, Behavioral Interaction, Cultural Characteristics, and Aesthetic Activity. The first two core categories pointed to the physical elements that were preconditions for forming HLIs; the remaining three pointed to the metaphysical elements that represented the distinctive characteristics of the HLIs. We proposed a structural model of HLIs comprising these five core categories from these findings (Figure 6). The word frequencies for the different attractions (Figure 7) revealed that the richness of the HLIs was positively proportional to word frequency.

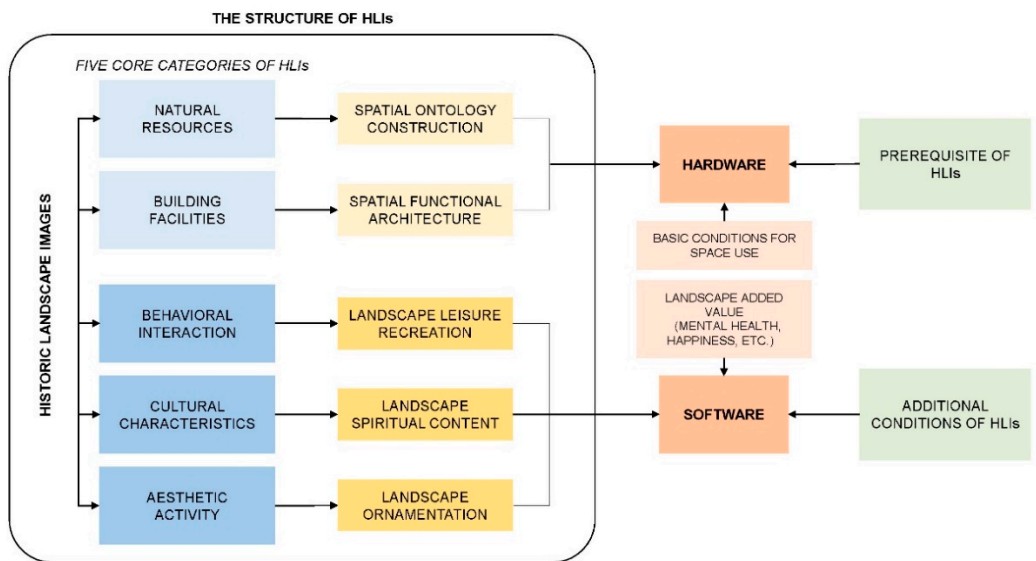

**Figure 6.** Structural model of the HLIs.

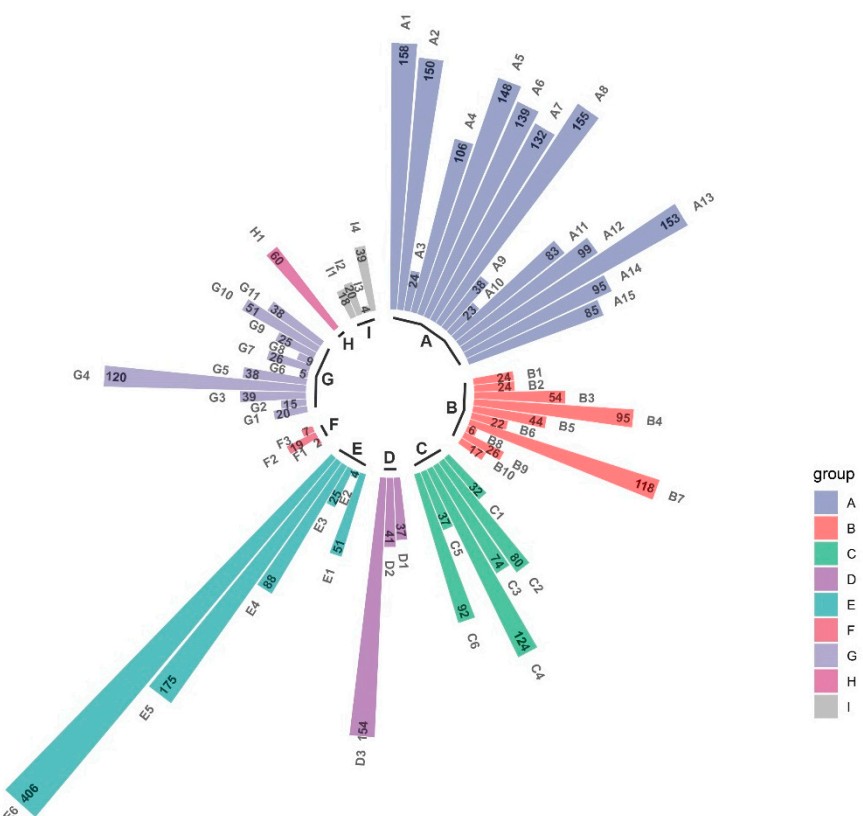

**Figure 7.** The relative association of different HLIs with the West Lake attractions.

## 3.2. Visitor Distribution

The GIS visualization based on the GPS data revealed the tourists' density distribution in West Lake, allowing us to examine their behaviors. We made a precise visualization of the GPS data (Figure 8) that showed that the GPS points almost covered the attractions of West Lake. However, the density distribution was uneven, with pronounced hot and cold spots. The GPS point and line density analyses showed roughly the same hot and cold spots.

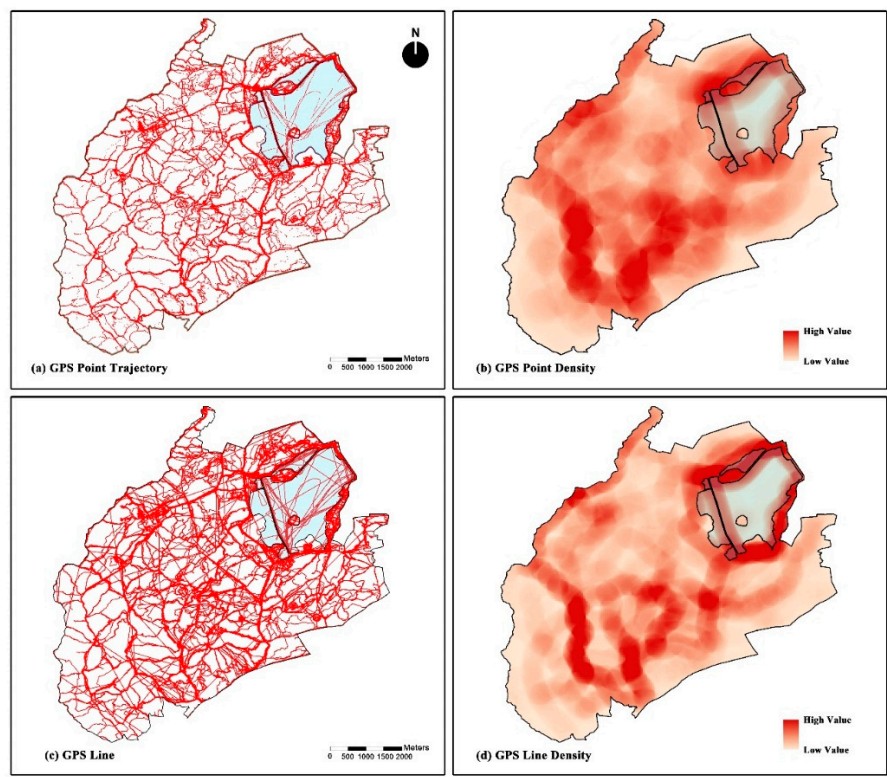

**Figure 8.** GPS visualization.

We generated another GPS kernel density map of West Lake (Figure 9). The ten attractions with the highest density were A1, H1, A11, A18, A2, A5, A10, C4, G5, and B7. Six of these were associated with the scenic area designated Group A. This finding confirmed that the attractions near the lake had more visits and were more popular among tourists.

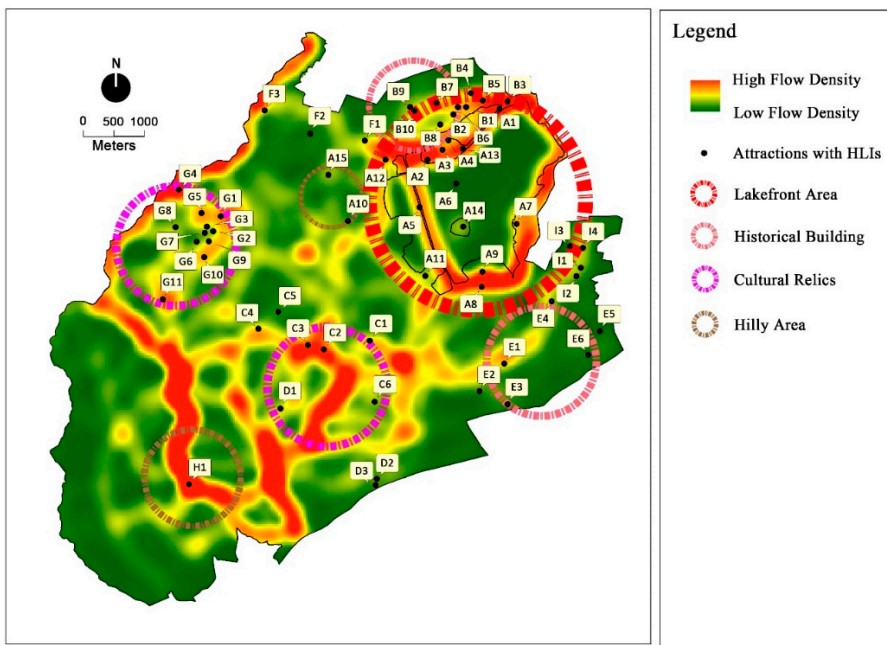

**Figure 9.** West Lake GPS dwell density.

### 3.3. Correlations between IATBs and HLIs

We calculated the number of visitor paths, visitor dwell time, average visitor dwell time, and tour speed within each attraction to determine whether IATB was affected by the HLIs. We conducted four regression analyses with HLIs as the independent variable, with the number of visitor paths, visitor dwell time, average visitor dwell time, and tour speed as the dependent variables.

#### 3.3.1. Visitor Paths

Using HLIs ($x$) as the independent variable and the number of visitor paths ($y_1$) as the dependent variable for linear regression analysis, we obtained the model equation as follows:

$$y_1 = 24.581 + 0.294\,x \tag{3}$$

The model R-squared value was 0.168; HLIs explained 16.8% of the variation in the number of tourist tour paths. The model passed the F-test (F = 11.529, $p = 0.001 < 0.01$), indicating that the HLIs significantly, positively influenced the visitors' path selection.

#### 3.3.2. Visitor Dwell Time

Visitor dwell time refers to the areas where visitors tarry the longest. These high-value areas included A1, A13, A2, A5, H1, and G5. Using HLIs ($x$) as the independent variable and tourist dwell time in minutes ($y_2$) as the dependent variable for linear regression analysis, we obtained the model equation as follows:

$$y_2 = 38.247 + 0.658\,x \tag{4}$$

The model R-squared value was 0.132. The HLIs explained 13.2% of the variation in visitor dwell time. The model passed the F-test (F = 8.701, $p = 0.005 < 0.01$), indicating that the HLIs significantly, positively influenced visitor dwell time.

#### 3.3.3. Average Visitor Dwell Time

To eliminate the effect of the number of visitors on dwell time, we calculated the average visitor dwell time to identify the places where visitors tarried the longest. We used HLIs ($x$) as the independent variable and the average visitor dwell time in seconds ($y_3$) as the dependent variable for linear regression analysis, obtaining the following model equation:

$$y_3 = 8.207 - 0.003\,x \tag{5}$$

The model did not pass the F-test (F = 0.172, $p = 0.679 > 0.01$), indicating that the HLIs did not have a meaningful relationship with the average length of visitors' stays and the correlation between the two was weak.

#### 3.3.4. Tour Speed

Using HLIs ($x$) as the independent variable and tour speed ($y_4$) as the dependent variable for linear regression analysis, we obtained the model equation as follows:

$$y_4 = 1.357 + 0.004\,x \tag{6}$$

The model did not pass the F-test (F = 0.256, $p = 0.615 > 0.01$), indicating that the HLIs did not have a meaningful relationship with tour speed and the correlation between the two was weak.

The results of the four regressions showed that the HLIs had a significant, positive effect on the number of visitor paths and visitor dwell time (Figure 10), but no significant effect on average visitor dwell time or tour speed.

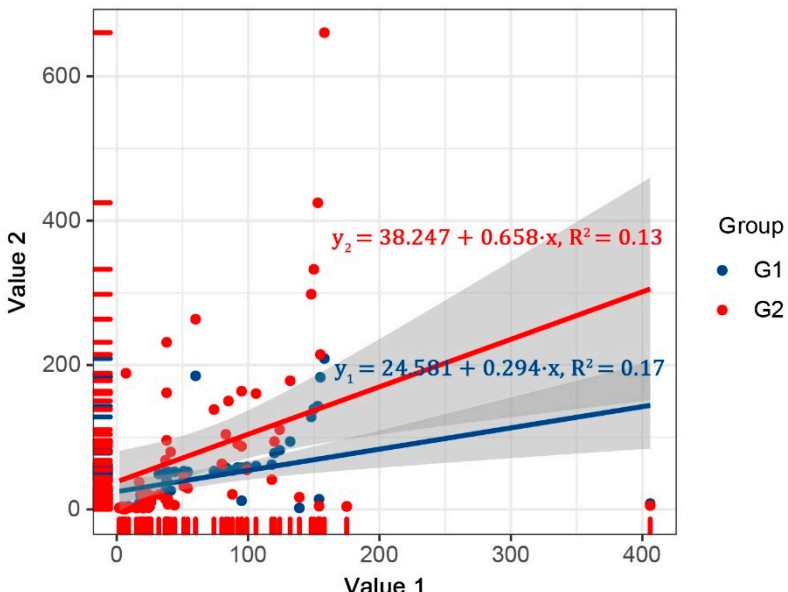

**Figure 10.** Model of the relationship between the number of visitor paths (**G1**), visitor dwell time (**G2**), and the HLIs.

We calculated the number of visitor paths for each West Lake attraction (Figure 11). The lakefront area had the most visitor paths and the densest visitor flow. The attractions with historical buildings and cultural relics had more visitor paths and a more dispersed visitor flow. The hilly area had the fewest visitor paths. In general, the tourists' travel behaviors were heterogeneous. The tourists usually chose only one route and could visit all the attractions because of West Lake's size. Therefore, they generally tarried longer at the most popular attractions. Though a flow of people passed through the cold spots, their visit times were relatively short and scattered, indicating that they considered those attractions less appealing.

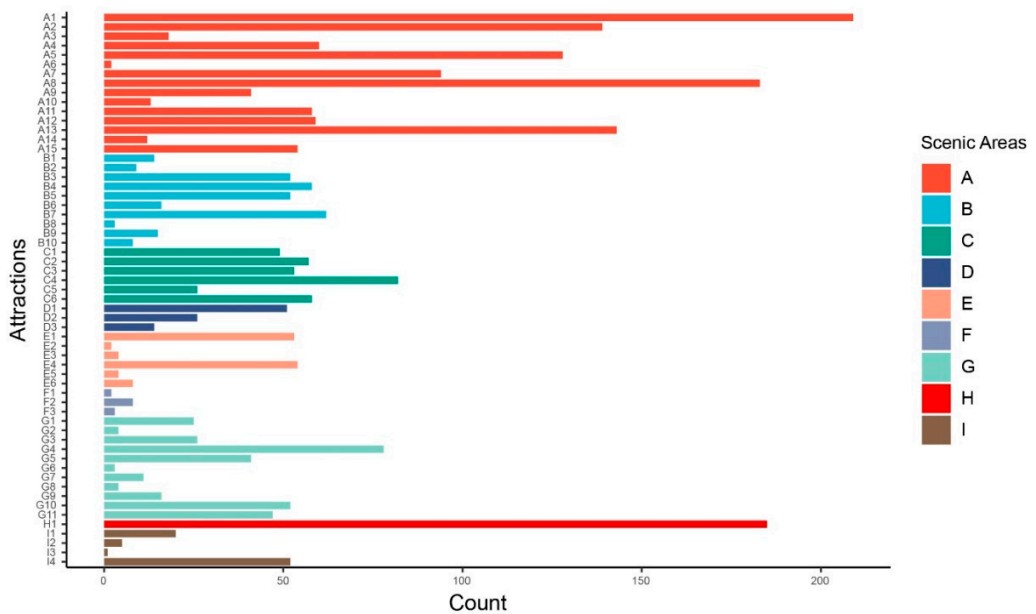

**Figure 11.** The number of visitor paths for each West Lake attraction.

We drew a scatterplot of the number of visitor paths among the West Lake attractions (Figure 12). We found some interesting features when we combined that with information

such as the dwell density. The scatterplot distribution can be divided into three kinds of clusters. The first (I) is high IATBs with high HLIs, the crucial feature observed by this study. The second (II) is low IATBs with high HLIs, including the attractions D3, E5, E6, A6, and A14. In ancient times, the first three of these attractions were important political and economic sites; however, they are no longer politically significant, and their management and maintenance are insufficient, thus, people rarely visit them now. The last two attractions are located in the middle of the lake, making them difficult to detect by GPS; therefore, their visitors' spatio-temporal behaviors were not fully recorded. The third kind of cluster (III) is high IATBs with low HLIs, as demonstrated by attraction H1, which is on a major traffic route in a mountainous area; therefore, many tourists pass through it.

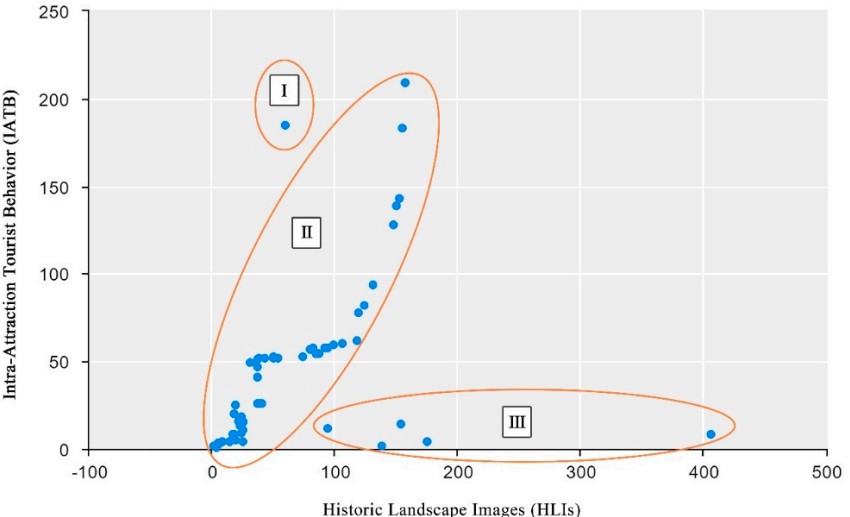

**Figure 12.** Scatterplot of the number of visitor paths in West Lake attractions.

## 4. Discussion

### 4.1. Influence of HLIs on IATBs

The combination of ancient poems and other literary works with landscapes is not accidental; it is a successful, time-tested form of aesthetic interaction [42]. The descriptions of the landscape images in ancient poems and the emotional reactions and reflections of the researchers during their visits provided suitable materials for analyzing the scenic images. Our study took a qualitative approach, applying grounded theory to the analysis of poems about the heritage site, constructing a structural model of HLIs through a three-level data coding process with academic significance.

Our study examined how HLIs influence IATBs. Previous studies have mainly focused on time, speed, and season as explanations for different spatio-temporal behaviors among visitors [36]; however, our findings suggested that HLIs can be a powerful and meaningful predictor of such behavior. Gaining a more robust understanding of IATBs could help improve tourism. Our study revealed uneven patterns in the West Lake tourists' spatio-temporal behaviors. The differences in their behaviors for the different attractions suggest that site managers should not promote all the attractions with an equal amount of effort. Using HLIs as an indicator, combined with GPS tracking data that accurately displays visitors' spatio-temporal behaviors, offers a new perspective that could further our understanding of IATBs at modern attractions and help site managers and administrators make targeted site-development recommendations.

The lakefront area of the West Lake heritage site had the highest HLIs and the most visitors. That highlights the importance of expanding the lakefront rest area and its infrastructure while ensuring that the natural waters are not polluted, and the beautiful scenery is maintained. The historical buildings and cultural relics around the lake have a moderate level of HLIs that make them attractive to tourists, easing the crowded situation at the lakefront

attractions. The HLIs are lower in the hilly area. We recommend adding attractive service facilities specific to the mountains for these less-crowded attractions to encourage more people to visit them, reducing the burden on the more popular attractions. In addition to the HLIs, we found a link between visitor behaviors and the major roads. Therefore, management needs to understand the distribution of routes and enhance the marketing of attractions around the major roads. The site managers could further strengthen their site management and promote sustainable development of future tourism by emphasizing HLIs.

### 4.2. HLIs Are an Influential Predictor of Future Tourism Trends for Heritage

Exploring tourists' spatio-temporal behavior with emerging modern landscape characteristics is common. However, this study suggests that visitor behaviors also depend on historical factors. We propose using HLIs to predict future tourism trends for heritage sites, creating a bridge to the future that can inform management targeting, tourism development, and promotion decisions. Managers can use HLIs as a theme for tourism marketing to create a poetic city image that blends the old and the new. Linking a city's image with the desire to "assert the future" can attract even more tourists, stimulate cultural consumption, and promote urban development. Creating and marketing new images for cities can improve their competitive position in attracting or retaining resources [43]. This could help Hangzhou compete more effectively with China's other "new first-tier cities" (e.g., Nanjing, Chengdu). This approach could be applied to other regions with HLIs and provide managers with meaningful lessons for urban planning and marketing to enhance their cities' images.

### 4.3. Limitations and Future Research

One, this study focused on the West Lake Heritage Site in Hangzhou, China, which offers both natural and cultural attractions in a unique landscape. Our results might differ from those for other heritage sites featuring different elements. Future research should examine the differences in visitor behaviors across multiple types and scales of destinations over a long period. Two, we took the HLIs we constructed for our study directly from ancient Chinese poetry; these could be considered in combination with other elements in the future. Digital access to ancient poems could increase the sample size. Three, uniform encoding might have oversimplified the data. For example, we uniformly replaced the words "flower", "grass", "willow", "peach", "laurel", "cypress", and "lotus" with the word "plant". This simplification might have made the attribute analysis of attractions flawed. Ideally, the coding should be finer grained. Four, some of the data in our study, such as the GPS data from the central lake, were difficult to obtain. Thus, the visitors' spatio-temporal behaviors might not have been wholly recorded, partially affecting the study's results. Five, the promotion of the "Ten Poetically Named Scenic Places", on websites such as Instagram and Weibo might have led to an increase in the number of visitors to those specific West Lake attractions, partially affecting the study's results. User generated content (UGC), such as that found on Weibo and Instagram, promotes attractions, which in turn influences visitor behavior. UGC recommendations for attractions follow certain principles. For example, they highlight attractions with good historical and cultural views. We believe that these attractions are accompanied by a high HLI. On this basis, HLIs have limitations, although they can still explain a part of tourists' behavior. UGC provides tourists with basic factual information about tourism products and destinations, increasing their understanding of the destination [44]. Additionally, UGC affects the willingness to choose a destination, consumer perceptions and travel decisions [45], and tourist loyalty [46]. The consideration of UGC can be increased in the future by conducting a multifactor regression analysis. Six, it was not possible to collect information on the tourists' emotions through open GPS track data, which limited the identification of causality and the further classification of tourists [47]. Future studies should select or configure data combined from multiple sources. Photo-taking behaviors often accompany open trajectory data. Future studies could use geo-tagged photos to determine well-being and mental health.

## 5. Conclusions

This study constructed a concept of HLIs by extracting images from ancient poems to understand the differences in visitors' IATBs at the West Lake Heritage Site. Our structural model of HLIs comprised five core categories: Natural Resources, Building Facilities, Behavioral Interaction, Cultural Characteristics, and Aesthetic Activity. We designed an extraction-storage scheme to collect GPS trajectory information about the visitors' IATBs: visitor paths, visitor dwell time, average visitor dwell time, and tour speed. We found that the visitor distribution was extremely uneven, and different attractions had different visitor behaviors. The GIS analysis also identified West Lake's cold and hot spots, indicating which attractions were most popular; we offered recommendations for these spots. The various attractions' locations within the immense park led to extreme behavioral variability, with high visitation in the lake area, medium in the crossover area, and low in the mountainous areas.

Our exploration of the significance of historical factors for modern landscapes presents a framework for a holistic approach to heritage site management and tourism in general. We linked theory and practice using images to study historic landscapes, a reverse process to capture features of interest, and an open framework for processing GPS data to establish correlations between HLIs and IATBs. Our findings should inform future heritage site management—and neighboring cities—about the benefits of using HLIs to predict attraction visitors' behaviors and leveraging those insights to optimize multiple-attraction sites proportionally. Such projections can provide new perspectives for heritage studies, landscape planning, and tourism image-making.

**Author Contributions:** Conceptualization, Q.L. and X.T.; methodology, Q.L.; software, Q.L. and K.L.; validation, Q.L., X.T. and K.L.; formal analysis, Q.L. and K.L.; investigation, Q.L. and K.L.; resources, X.T.; data curation, Q.L.; writing—original draft preparation, Q.L., X.T. and K.L.; writing—review and editing, Q.L.; visualization, Q.L.; supervision, X.T.; project administration, Q.L. and X.T.; funding acquisition, Q.L. and X.T. All authors have read and agreed to the published version of the manuscript.

**Funding:** This research was funded by Postgraduate Research & Practice Innovation Program of Jiangsu Province, Grant Number KYCX22_1102 (China); Priority Academic Program Development of Jiangsu Higher Education Institutions, Grant Number 164120281 (China); Humanities and Social Sciences Research Planning Project, Ministry of Education, Grant Number 22YJA760075 (China); National Forestry and Grassland Administration, Grant Number 2021LYYB02 (China).

**Institutional Review Board Statement:** The appropriate ethics review board approved the study design.

**Informed Consent Statement:** Informed consent was obtained from all subjects involved in the study.

**Data Availability Statement:** The data that support the findings of this study are available upon reasonable request from the authors.

**Conflicts of Interest:** The authors declare no conflict of interest.

## Appendix A

**Table A1.** Fifty-nine attractions with HLIs in West Lake.

| Serial Number | Scenic Areas | Scenic Area Number | Attractions Number | Attractions |
|---|---|---|---|---|
| [1] | | | A1 | Lingering Snow on Broken Bridge |
| [2] | | | A2 | Xiling Printing House |
| [3] | | | A3 | Sizhao Pavilion |
| [4] | | | A4 | Crane in Plum |
| [5] | | | A5 | Su Causeway in the Morning of Spring |
| [6] | | | A6 | Mid-Lake Pavilion |
| [7] | | | A7 | Orioles Singing in the Willows |
| [8] | Lakefront | A | A8 | Evening Bell Ringing at Nanping Hill |

**Table A1.** *Cont.*

| Serial Number | Scenic Areas | Scenic Area Number | Attractions Number | Attractions |
|---|---|---|---|---|
| [9] | | | A9 | Leifeng Pagoda in Evening Glow |
| [10] | | | A10 | MAO's Home Port |
| [11] | | | A11 | Viewing Fish at Flowery Pond |
| [12] | | | A12 | Breeze-ruffled Lotus at Winding Garden |
| [13] | | | A13 | Autumn Moon over the Calm Lake |
| [14] | | | A14 | Three Pools Mirroring the Moon |
| [15] | | | A15 | Twin Peaks Piercing the Cloud |
| [16] | | | B1 | Zhiguo Temple |
| [17] | | | B2 | Manao Temple |
| [18] | | | B3 | Wanghu Building |
| [19] | | | B4 | Precious Stone Hill Floating in Rosy Cloud |
| [20] | North Hill | B | B5 | Precious Stone Hill Statue |
| [21] | | | B6 | Baoyun Temple |
| [22] | | | B7 | Ge Ling Chao |
| [23] | | | B8 | Site of Zhaolan Temple |
| [24] | | | B9 | Purple Cloud Cave |
| [25] | | | B10 | Tomb of Niuhao |
| [26] | | | C1 | Stone House Cave |
| [27] | | | C2 | Water Music Cave |
| [28] | Tiger Spring and Longjing Well | C | C3 | Haze at Sunset Cave |
| [29] | | | C4 | Longjing Well with Tea |
| [30] | | | C5 | Eight Sights of Longjing Well |
| [31] | | | C6 | Tiger Spring |
| [32] | | | D1 | Nine Creeks and Eighteen Gullies |
| [33] | Qiantang River | D | D2 | Liuhe Pagoda |
| [34] | | | D3 | Nine Streams to Watch The Tide |
| [35] | | | E1 | Ci Yunling Statues |
| [36] | | | E2 | Denon Temple |
| [37] | Phoenix Hill | E | E3 | Tomb of Wuhanyue |
| [38] | | | E4 | Myriad Pines Academy |
| [39] | | | E5 | Southern Song Dynasty Imperial City Site |
| [40] | | | E6 | Fantian Temple Heritage Park |
| [41] | | | F1 | Zhangxian Tomb |
| [42] | Botanical Garden | F | F2 | Jade Spring |
| [43] | | | F3 | Appreciate Plum Blossoms in Lingfeng Hill |
| [44] | | | G1 | Clock Sinking in Hejian River |
| [45] | | | G2 | Peak That Flew from Afar |
| [46] | | | G3 | Cold Spring Pavilion |
| [47] | | | G4 | North Peak |
| [48] | | | G5 | Lingyin Temple |
| [49] | Lingzhu | G | G6 | Mengxie Pavilion |
| [50] | | | G7 | Cuiwei Pavilion |
| [51] | | | G8 | Yongfu Temple |
| [52] | | | G9 | Stone with Three Lifetimes |
| [53] | | | G10 | Fajing Temple |
| [54] | | | G11 | Faxi Temple |
| [55] | Color Clouds | H | H1 | Color Clouds Hill |
| [56] | | | I1 | Three-Thatched-Hut Temple |
| [57] | Wu Mountain | I | I2 | Clouds Living and Pine's Sound |
| [58] | | | I3 | Chenghuang Temple |
| [59] | | | I4 | Qingyi Cave |

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
