# Peer review of "Do Historic Landscape Images Predict Tourists’ Spatio-Temporal Behavior at Heritage Sites? A Case Study of West Lake in Hangzhou, China"

_land, doi:10.3390/land11101643_

Round 1
Reviewer 1 Report
This paper studies the connection between historic landscape images and intra-attraction tourist behaviour using both qualitative content analysis and the data derived from GPS technology and tries to find out spatiotemporal behavioural tourist activities. The findings suggest that image and perception of the heritage site has a direct influence over visitor behaviour as the destination. Some interesting managerial implications are also discussed. Except for some typos and minor grammatical errors, paper overall is well written and easy to follow. When it comes to studying visitor behaviour, some data triangulation through non-participant observation would have been useful in ascertaining the correlation more. My only concern is since the data is largely all gathered through unobtrusive method, we do not know if there are specific reason that visitors spend time some places that other.
Author Response
Response to Reviewer 1 Comments
Dear Editors and Reviewer:
Thank you for your letter and for the reviewer’s comments concerning our manuscript entitled “Do historic landscape images predict tourists’ spatiotemporal behavior in heritage sites? A case study of West Lake in Hangzhou, China” (ID: land-1914004). Those comments are all valuable and very helpful for revising and improving our paper. We have studied comments carefully and have made corrections which we hope meet with approval. I have sent my paper to a professional English editing service and received a certificate. All changes are marked in the revised version of the manuscript using the “Track Changes” function. The main corrections in the paper and the responds to the reviewer's comments are as flowing:
Point: My only concern is since the data is largely all gathered through unobtrusive method, we do not know if there are specific reason that visitors spend time some places that other.
Response:
Thank you for your kind suggestions.
We collect generalized trajectory data. Reduce errors in the analysis of visitor behavior due to data collection. The trajectory we collect includes both individual and organized trips, and we do not collect trajectories based on a specific population profile. Different types of visit for different purposes, some to enjoy the scenery and some to get fit. Therefore, the probability of crossing each area is equal. The data we collect is available 24 hours a day, and no specific time of day is collected for a particular period of visit. Most of the data uploaded by users are during the daytime. We did not limit the length of the visit, and most of the data collected had a relatively ample visit time. Each attraction is well described in brochures, printed guides, and online guides. Roads are marked with signposts, directions and information signs, so there is a wealth of information to guide the walk to each attraction.
However, visitor behavior may still be influenced by other factors, such as traffic reasons, political and economic reasons, as described in the conclusion of the article. In addition, the website can also be influenced by websites such as Instagram and Weibo. They will highlight the "Ten Poetically Named Scenic Places" located in the lakefront area. These attractions have a long history and are considered by UNESCO to be the most important attractions in West Lake. The promotion of these sites may have a role in guiding visitors. We have included this in the limitations. Please see lines 484-486 for details.
Overall, we found that the correlation between HLIs and IATB is non-negligible through data analysis. Tourists' behavior is influenced by HLIs.
We tried our best to improve the manuscript and made some changes in the manuscript.
We appreciate for Editors/Reviewers' warm work earnestly, and hope that the correction will meet with approval. Once again thank you very much for your comments and suggestions.

Reviewer 2 Report
land-1914004-peer-review-v1
Review of Do historic landscape images predict tourists’ spatiotemporal behavior in heritage sites? A case study of West Lake in Hangzhou, China
Methodology
Line 152 give reference for and briefly describe functionality of ROST CM6 software. This cannot be assumed for international audiences
Line 159 “and other sources” these should be listed specified in an appendix
Line 162 Why were only Two-thirds of the texts screened to correct obvious errors and not all texts? Even if the remaining third is used for theoretical saturation tests it still needs to be screened for errors…explain
Line 174-75 I am concerned that this uniform coding simplifies the data in an excessively reductionist fashion. When it comes to attractions, ‘flower’ and ‘lotus’ are very different from ‘grass’ I am concerned that this reduction may have flawed the attribute analysis of the attractions. Ideally the coding should be more fine grained. If that cannot be done, then the limitations need to be clearly discussed
Lines 181–185 These sites need more discussion. What kind of data could be extracted? This cannot be assumed for international audiences.
I am concerned that the authors have not addressed any ethical considerations derived from this. Just because tourists place their paths onto a website does not imply that they provide informed consent that their data points are being used to analyses their behaviours. Ethics approval (or formal waiver) will be required here.
On a fundamental level I am missing a discussion of several parameters that may influence the outcomes of the study
1) What is the travel behaviour of the individuals whose tracks have been harvested for the data set. Are they individual travellers or did they travel as part of an organised tour? Tours tend to emphasise specific points and either quickly traverse ‘uninteresting’ areas or not pass through them at all.
2) What level of formal promotion / tourist information exists in terms of brochures, printed guides and online guides. Are there signposts, direction and information signs on the paths?
3) Which of the attractions are highly publicised on Weibo? It is well known that attractions that are publicised on sites such as Instagram and Weibo see disproportionate visitation compared to sites that are not promoted in that way.
4) What is the duration of the visits. People who are time poor will visit only for short time and will limit what they see compared to people with more leisure
Minor issues
Some of the tables are broken across pages
Ethics
The study collected user movement data , but there is no comment on formal approval of the study by the human ethics review board of the authors’ institution(s). In this day and age this must be provided. If formal approval was not sought or obtained, the paper must be rejected out of hand as the credibility of the journal is at stake.
Author Response
Response to Reviewer 2 Comments
Dear Editors and Reviewer:
Thank you for your letter and for the reviewer’s comments concerning our manuscript entitled “Do historic landscape images predict tourists’ spatiotemporal behavior in heritage sites? A case study of West Lake in Hangzhou, China” (ID: land-1914004). Those comments are all valuable and very helpful for revising and improving our paper. We have studied comments carefully and have made corrections which we hope meet with approval. I have sent my paper to a professional English editing service and received a certificate. All changes are marked in the revised version of the manuscript using the “Track Changes” function. The main corrections in the paper and the responds to the reviewer's comments are as flowing:
- Point 1: Line 152. Give reference for and briefly describe functionality of ROST CM6 software. This cannot be assumed for international audiences.
Response 1: Your comments are thoughtful. We have added a description of the ROST CM6.0 software in lines 180-182 and provided references to previous studies using ROST CM6.0.
- Point 2: Line 159. “and other sources”, these should be listed specified in an appendix.
Response 2: We are very sorry for our incorrect writing of “other sources”. We have deleted “and other sources” phrases carefully. The details are in lines 165. Please kindly review.
- Point 3: Line 162. Why were only Two-thirds of the texts screened to correct obvious errors and not all texts? Even if the remaining third is used for theoretical saturation tests it still needs to be screened for errors.
Response 3: We apologize for the errors in our writing here. We corrected all the texts for obvious errors (content recording errors, punctuation errors, spelling errors, etc.) and deleted duplicate and irrelevant content. We collected 1,131 poems, extracted 3,993 landscape image terms, and then randomly selected two-thirds of the sample for coding analysis and model construction. We reserved the remaining one-third of the sample for theoretical saturation tests. The details are in lines 168-172.
- Point 4: Line 174-75. I am concerned that this uniform coding simplifies the data in an excessively reductionist fashion. When it comes to attractions, ‘flower’ and ‘lotus’ are very different from ‘grass’ I am concerned that this reduction may have flawed the attribute analysis of the attractions. Ideally the coding should be more fine grained. If that cannot be done, then the limitations need to be clearly discussed.
Response 4: Thank you for your kind suggestions. In our coding analysis, flowers and grasses are subordinated to the categories belonging to plants without more specific subdivision, which has certain limitations. The details are in lines 477-481.
- Point 5: Lines 181–185. These sites need more discussion. What kind of data could be extracted? This cannot be assumed for international audiences. I am concerned that the authors have not addressed any ethical considerations derived from this. Just because tourists place their paths onto a website does not imply that they provide informed consent that their data points are being used to analyses their behaviours. Ethics approval (or formal waiver) will be required here.
Response 5: Thank you for your kind reminders. Our data acquisition has been licensed from two websites. We have collected GPS tracking data from 2bulu.com and foooooot.com, the leading outdoor resource-sharing sites in China. Users have signed a consent to public use agreement when registering, agreeing to share trails for public release and download for use. If the user does not want to disclose the track data, it will be visible only to the user and cannot be downloaded and used by others. The data we use is the data that the user has agreed to use publicly, not private data, which is in line with ethical standards. We have added an ethics statement to line 197-198.
- Point 6: What is the travel behaviour of the individuals whose tracks have been harvested for the data set. Are they individual travellers or did they travel as part of an organised tour? Tours tend to emphasise specific points and either quickly traverse ‘uninteresting’ areas or not pass through them at all.
Response 6:The track data we collect encompasses both individual and organized trips, and no tracks are collected based on a particular population profile. Different types of tourists visit for different purposes, some for scenery and some for fitness. Therefore, we consider the probability of crossing each area to be equal.
- Point 7: What level of formal promotion/tourist information exists in terms of brochures, printed guides and online guides. Are there signposts, direction and information signs on the paths?
Response 7: Each site is well described in brochures, printed guides and online guides. The location and brief description of each attraction are described. Roads are marked with signposts, directions and information signs, so there is a wealth of information to guide visitors to each attraction.
- Point 8: Which of the attractions are highly publicised on Weibo? It is well known that attractions that are publicised on sites such as Instagram and Weibo see disproportionate visitation compared to sites that are not promoted in that way.
Response 8: Websites such as Instagram and Weibo will highlight the "Ten Poetically Named Scenic Places" located in the lakefront area. These attractions have a long history and are considered by UNESCO to be the most important attractions in West Lake. The promotion of these sites may have a role in guiding visitors. We have included this in the limitations. Please see lines 484-486 for details.
- Point 9: What is the duration of the visits. People who are time poor will visit only for short time and will limit what they see compared to people with more leisure
Response 9: The data we collect is available 24 hours a day and there is no specific time period for collecting visits. Most of the data uploaded by users are during the daytime. We do not limit the time of visit. Most of the data paths we collect have a relatively long visit time.
- Point 10: Some of the tables are broken across pages.
Response 10: We have reformatted the tables. Please kindly review. Thank you.
Ethics
The appropriate ethics review board approved the study design. We only collect data that users have agreed to share publicly and do not use their private data in accordance with ethical and moral standards.
We tried our best to improve the manuscript and made some changes in the manuscript.
We appreciate for Editors/Reviewers' warm work earnestly, and hope that the correction will meet with approval. Once again thank you very much for your comments and suggestions.

Round 2
Reviewer 2 Report
Review of land-1914004-peer-review-v2
The authors have adequately addressed most aspects of the first review and are to be commended. However, some still require further attention. As it stands, these shortcomings are fatal flaws
Initial review |
Point 7: What level of formal promotion/tourist information exists in terms of brochures, printed guides and online guides. Are there signposts, direction and information signs on the paths? |
Author’s response |
Response 7: Each site is well described in brochures, printed guides and online guides. The location and brief description of each attraction are described. Roads are marked with signposts, directions and information signs, so there is a wealth of information to guide visitors to each attraction. |
Further action needed |
Thank you. But HOW do these sign postings influence the user behaviour at the site? User behaviour is not random, but guided by prior information and by signposting at the site. This MUST be discussed as the signposting may well introduce a major bias |
Initial review |
Point 8: Which of the attractions are highly publicised on Weibo? It is well known that attractions that are publicised on sites such as Instagram and Weibo see disproportionate visitation compared to sites that are not promoted in that way. |
Author’s response |
Response 8: Websites such as Instagram and Weibo will highlight the "Ten Poetically Named Scenic Places" located in the lakefront area. These attractions have a long history and are considered by UNESCO to be the most important attractions in West Lake. The promotion of these sites may have a role in guiding visitors. We have included this in the limitations. Please see lines 484-486 for details |
Further action needed |
But HOW do the promotion of these sites on Instagram and Weibo, which is SELECTIVE in what is shown, influence the user behaviour at the site? User behaviour is not random, but guided by prior information and by signposting at the site. This MUST be discussed as the signposting may well introduce a major bias. Looking at the limitations section, this is NOT discussed despite your assertions |
Initial review |
Point 9: What is the duration of the visits. People who are time poor will visit only for short time and will limit what they see compared to people with more leisure |
Author’s response |
Response 9: The data we collect is available 24 hours a day and there is no specific time period for collecting visits. Most of the data uploaded by users are during the daytime. We do not limit the time of visit. Most of the data paths we collect have a relatively long visit time. |
Further action needed |
This does not answer my question. What is the duration of the visits that were analysed. Surely short duration visits will limit what they look at which long duration visits will allow people to see more things. This MUST be discussed |
Author Response
Response to Reviewer 2 Comments
Dear Editors and Reviewer:
Thank you for your letter and for the reviewer’s comments concerning our manuscript entitled “Do historic landscape images predict tourists’ spatiotemporal behavior in heritage sites? A case study of West Lake in Hangzhou, China” (ID: land-1914004). Those comments are all valuable and very helpful for revising and improving our paper. We have studied comments carefully and have made corrections which we hope meet with approval. I have sent my paper to a professional English editing service and received a certificate. All changes are marked in the revised version of the manuscript using the “Track Changes” function. The main corrections in the paper and the responds to the reviewer's comments are as flowing:
- Point 1: HOW do these sign postings influence the user behaviour at the site? User behaviour is not random, but guided by prior information and by signposting at the site. This MUST be discussed as the signposting may well introduce a major bias.
Response 1: Thank you for your kind suggestions. We found signposts and brochures for West Lake. Each signpost will give directions to nearby attractions, which we think is just to guide the traffic. The brochures will introduce the name of each attraction, not just a few, without focus, also to guide the traffic. So we don't think this causes major bias. The prior information you mentioned we think is usually led by online UGC content, we write it in response2. Please kindly review.
- Point 2: But How do the promotion of these sites on Instagram and Weibo, which is SELECTIVE in what is shown, influence the user behaviour at the site? User behaviour is not random, but guided by prior information and by signposting at the site. This MUST be discussed as the signposting may well introduce a major bias. Looking at the limitations section, this is NOT discussed despite your assertions.
Response 2: Your comments are thoughtful. We further discussed the limitations of using user generated content (UGC) and future research. User generated content (UGC), such as Weibo and Instagram, promotes attractions, which in turn influences visitor behavior. UGC recommendations for attractions follow certain principles. For example, they highlight attractions with good historical and cultural views. We believe that these attractions are accompanied by high HLI. on this basis, HLI has limitations although it can still explain a part of tourists' behavior. UGC provides tourists with basic factual information about tourism products and destinations, increasing their understanding of the destination. Also, UGC affects willingness to choose a destination, consumer perceptions and travel decisions and tourist loyalty. The consideration of UGC can be increased in the future by conducting multi-factor regression analysis. The details are in lines 484-495.
- Point 3: What is the duration of the visits that were analysed. Surely short duration visits will limit what they look at which long duration visits will allow people to see more things. This MUST be discussed.
Response 3: Thank you for your kind reminders. The shortest trajectory we collected was 1 hour and 52 minutes, and the longest was 16 hours and 16 minutes. Regardless of the length of time, visitors will choose the sites they want to visit first. Short-time visitors would visit the attractions they wanted to visit and then leave directly, while long-time visitors would visit the attractions they wanted to visit first and then continue to visit other sites as they wished. This creates a gap in the frequency of attraction visits, but does not conflict with the findings of this paper. In our study, time was recorded in terms of GPS points, not total duration. Therefore, the total duration is not important in the study.
We tried our best to improve the manuscript and made some changes in the manuscript.
We appreciate for Editors/Reviewer's warm work earnestly, and hope that the correction will meet with approval. Once again thank you very much for your comments and suggestions.
